# Optimization of Rifapentine-Loaded Lipid Nanoparticles Using a Quality-by-Design Strategy

**DOI:** 10.3390/pharmaceutics12010075

**Published:** 2020-01-17

**Authors:** Joana Magalhães, Luise L. Chaves, Alexandre C. Vieira, Susana G. Santos, Marina Pinheiro, Salette Reis

**Affiliations:** 1LAQV, REQUIMTE, Departamento de Ciências Químicas, Faculdade de Farmácia, Universidade do Porto, 4050-313 Porto, Portugalluiselopes@gmail.com (L.L.C.);; 2i3S-Instituto de Investigação e Inovação em Saúde, INEB—Instituto de Engenharia Biomédica, Universidade do Porto, 4200-135 Porto, Portugal; 3ICBAS—Instituto de Ciências Biomédicas Abel Salazar, Universidade do Porto, 4050-313 Porto, Portugal

**Keywords:** drug delivery, infectious diseases, nanomedicine, lipid nanoparticles

## Abstract

This work aims to optimize and assess the potential use of lipid nanoparticles, namely nanostructured lipid carriers (NLCs), as drug delivery systems of rifapentine (RPT) for the treatment of tuberculosis (TB). A Box–Behnken design was used to increase drug encapsulation efficiency (EE) and loading capacity (LC) of RPT-loaded NLCs. The optimized nanoparticles were fully characterized, and their effect on cell viability was assessed. The quality-by-design approach allowed the optimization of RPT-loaded NLCs with improved EE and LC using the minimum of experiments. Analyses of variance were indicative of the validity of this model to optimize this nanodelivery system. The optimized NLCs had a mean diameter of 242 ± 9 nm, polydispersity index <0.2, and a highly negative zeta potential. EE values were higher than 80%, and differential scanning calorimetry analysis enabled the confirmation of the efficient encapsulation of RPT. Transmission electron microscopy analysis showed spherical nanoparticles, uniform in shape and diameter, with no visible aggregation. Stability studies indicated that NLCs were stable over time. No toxicity was observed in primary human macrophage viability for nanoparticles up to 1000 μg mL^−1^. Overall, the optimized NLCs are efficient carriers of RPT and should be considered for further testing as promising drug delivery systems to be used in TB treatment.

## 1. Introduction

Tuberculosis (TB) is the top infectious killer worldwide and one of the top 10 causes of deaths in 2018 [1]. The current TB treatment includes a standard six-month course of four antimicrobial drugs (i.e., rifampicin, ethambutol, isoniazid, and pyrazinamide) that, despite being effective, is associated with adverse effects [2,3]. The long treatment schedule that is required and the associated multiple side effects contribute to low patient compliance to therapy and to the appearance of multidrug-resistant strains [2,4]. Moreover, without proper treatment, up to two-thirds of people ill with TB will die [5].

The progress that has been made in combating TB should be intensified to eradicate the TB epidemic. The discovery and development of new effective strategies for TB therapy constitutes one of the key components for the “End TB strategy” proposed by the World Health Organization (WHO) [6]. In 2018, there were half a million new cases of rifampicin-resistant TB, of which 78% had multidrug-resistant TB [1]. Despite significant technological innovations introduced in the last 10 years, the multidrug-resistant TB crisis, undetected or unnotified TB cases, a suboptimal response to the TB and HIV co-epidemic, the high costs for TB patients, and the slow uptake of new effective tools constitute persistent and serious challenges in tackling the TB epidemic [1,2,6]. This fact encourages the development of new treatment regimens including other anti-TB drugs.

Rifapentine (RPT) is a rifamycin derivative that has been synthesized to prolong half-life and to improve the antimycobacterial activity of rifampicin, the most effective first-line anti-TB drug [7,8]. In fact, RPT effectiveness in drug-susceptible TB treatment has been studied in three clinical trials (i.e., TB Trial Consortium (TBTC) study 31/ACTG A5349, TBTC study 35, and TBTC study 37) [1]. Despite the relevance of this drug, RPT has several adverse effects, including nausea, discoloration of body fluids (i.e., tears, saliva, sputum, and urine), and acute renal failure [9,10]. Thus, the development of a new delivery system using nanotechnology is required to improve RPT pharmacokinetics and pharmacodynamics, minimizing its side effects.

In the last decades, the application of nanotechnology to health sciences and, more specifically, to drug delivery has opened up new opportunities for treating several diseases [11,12]. Nanoparticles, namely lipid nanoparticles, are considered biodegradable, safe, and constitute a promising strategy for easier and shorter treatment regimens. In the last recent years, quality-by-design studies using response surface methodologies have been used to optimize several drug-loaded nanoformulations [13,14,15,16,17,18]. Accordingly, the present work proposes the optimization and development of nanostructured lipid carriers (NLCs) loaded with RPT as a new delivery system for TB treatment. For that purpose, a Box–Behnken design was applied to optimize the nanoformulations to have a higher drug encapsulation efficiency (EE) and loading capacity (LC). The nanoparticle size was also optimized to be suitable for different delivery systems, including the oral and pulmonary administrations routes. After optimization, nanoformulations were fully characterized, and their effect on primary human macrophage (main cell target of *Mycobacterium tuberculosis*) viability was tested to understand if these particles would be appropriate for more effective and economic treatment regimens.

## 2. Materials and Methods

### 2.1. Preparation of the Formulations

#### Nanoparticles Synthesis

NLCs were produced by hot ultra-sonication based on the method described by Magalhães et al. [19]. Briefly, the lipids glyceryl palmitostearate (Precirol^®^ATO5; Gattefosé, Lyon, France), caprylic/capric triglyceride (Miglyol^®^812; Acofarma, Madrid, Spain), and the surfactant polysorbate 80 (Tween^®^80; Merck, Darmstadt, Germany) were weighted and heated in a water bath up to 70 °C. When the solid lipid was fully melted, 6 mL of preheated (T = 70 °C) Milli-Q^®^ double-deionized water (conductivity less than 0.1 µS cm^−1^) was added to the lipid phase. This mixture was then homogenized using a probe-sonicator (Vibra-Cell model VCX 130; Sonics and Materials Inc., Newtown, CT, USA) with a tip diameter of 6 mm at 70% amplitude for 5 min. Nanoemulsions were left to cool down and stored at room temperature. RPT-loaded NLCs were prepared using the above-mentioned approach but adding 10 mg of RPT (Sigma-Aldrich GmbH, Hamburg, Germany) to the lipid phase before melting. After synthesis, the nanoformulations were stored at 20 °C during 3 months.

### 2.2. Experimental Design

#### 2.2.1. Box–Behnken Design

A 3-level, 3-factor Box–Behnken design was performed to select the most appropriate NLC composition content and to test the effect of independent variables on dependent variables, using the minimum of experiments. The established independent variables were the amount of solid lipid (*X*_1_), the amount of liquid lipid (*X*_2_), and the amount of surfactant (*X*_3_). Three different levels of each independent variable were tested (Table 1), testing lower and upper values for each variable based on pre-formulation studies and literature research. The studied dependent variables were the particle size (*Y*_1_), EE (*Y*_2_), and LC (*Y*_3_) (Table 1). The amount of drug (10 mg), sonication amplitude (70%), sonication time (5 min), and final volume (6 mL) were set at fixed levels.

Data were analyzed in STATISTICA 10 (Statsoft Power Solutions, Inc., Tulsa, OK, USA) software, using the following polynomial equation: *Y* = b_0_ + b_1_*X*_1_ + b_2_*X*_2_ + b_3_*X*_3_ + b_12_*X*_1_*X*_2_ + b_13_*X*_1_*X*_3_ + b_23_*X*_2_*X*_3_ + b_11_*X*_1_^2^ + b_22_*X*_2_^2^ + b_33_*X*_3_^2^.
where *Y* corresponds to the dependent variable; b_0_ is the intercept; *X*_1_, *X*_2_, and *X*_3_ are the coded levels of independent variables; and b_1_ to b_33_ are the regression coefficients computed from the observed experimental values of *Y*. The terms *X*_1_*X*_2_ and *Xi*^2^ (*i* = 1, 2, or 3) represent the interaction and quadratic terms, respectively. The result was statistically validated using ANOVA, by statistical significance of coefficients and *R*^2^ values. Statistical analyses were considered significant for *p* values < 0.05.

#### 2.2.2. Optimization and Validation

STATISTICA 10 (Statsoft Power Solutions, Inc., Tulsa, OK, USA) software was used to perform the graphical and numerical analyses of the optimum values of the dependent variables based on the desirable criteria (Table 1). To validate the experimental design, three NLC replicates were synthetized using the selected formulation composition indicated by the Box–Behnken design. The values obtained for the dependent variables were compared to the predicted values of the Box–Behnken design.

### 2.3. NLC Characterization

#### 2.3.1. Particle Size, Polydispersity, and Surface Charge

The hydrodynamic size distribution, polydispersity (PDI), and the surface charge (ζ-potential) of the developed formulations were characterized using dynamic and electrophoretic light scattering using a ZetaPALS/ZetaPotential Analyzer (Brookhaven Instruments, Holtsville, NY, USA), operating at a scattering angle of 90°, at 20 °C, with a dust cut-off set to 30. Prior to the measurements, formulations were diluted (1:100) in double-distilled water and filtered with a syringe filter (800 nm). For mean hydrodynamic diameter and PDI, six runs of 2 min were performed at each measurement. For ζ-potential determination, ten runs with ten cycles were performed at each measurement. All measurements were done in triplicate, and results were expressed as mean ± standard deviation (SD).

#### 2.3.2. Drug Encapsulation Efficiency and Loading Capacity

The EE of RPT within NLCs was determined by calculating the difference between the amount of RPT used to prepare the formulations and the amount of free RPT in the aqueous phase [20]. Briefly, a dilution (1:40) of RPT-loaded NLCs in double-deionized water was transferred to an Ultrafree^®^ Centrifugal Filter Device with a nominal molecular weight cut off of 50,000 (Burlington, MA, USA) and centrifuged at 3000× *g* for 10 min using a Jouan BR4i multifunction centrifuge with a KeyWrite-DTM interface (Thermo Electron, Waltham, MA, USA). The non-entrapped drug present in the supernatant was quantified at 337 nm using UV/Vis spectrophotometry (Jasco, Inc., Easton, MD, USA). A standard curve of RPT was used to determine RPT concentration. All measurements were done in triplicate. LC was defined by the ratio between the amount of drug encapsulated and the total lipid amount. All measurements were done in triplicate, and results were expressed as mean ± SD.

#### 2.3.3. Lyophilization

Lyophilization was performed based on the methods described by Varshosaz et al. [21] and Vieira et al. [22]. Briefly, the samples were prepared using aerosil 2% (w/w) (Sigma-Aldrich, MA, USA) as cryoprotectant. The process involved an initial freezing at −60 °C for 720 min, and a condensation process made at −80 °C, under 150 mTorr of pressure. The first drying was done at 20 °C for 1200 min, under 150 mTorr of pressure, and the secondary drying was at 25 °C, for 1200 min, under 100 mTorr of pressure. All the procedures were performed using a VirTis freeze dryer (Advantage Plus EL-85; SP Scientific, Gardiner, NY, USA). After lyophilization, the nanoformulations were stored at 20 °C during 3 months.

#### 2.3.4. Storage Stability

To evaluate the physico-chemical stability of the developed nanoparticles (i.e., lyophilized formulations and nanoparticles in liquid suspension), the hydrodynamic size distribution, PDI, ζ-potential, and EE were measured over time during 3 months. All these properties were measured according to the procedures mentioned in the previous sections (i.e., Section 2.3.1 and Section 2.3.2).

#### 2.3.5. Differential Scanning Calorimetry Analysis

Differential scanning calorimetry (DSC) measurements were performed to study the structural properties of the developed nanoformulations. Approximately 5 mg of each formulation was weighted in aluminum pans that were hermetically sealed. Samples were heated at a constant heating rate of 10 °C min^−1^ from 10 to 200 °C in an inert atmosphere, maintained by purging nitrogen gas at a flow rate of 50 mL min^−1^. The measurements were performed using a differential scanning calorimeter DSC 200 F3 Maia (Netzsch Group, Selb, Germany). An empty aluminum pan was used as reference. The onset, melting point (peak maximum), and melting enthalpy (ΔH) were calculated using the software provided by the DSC equipment (NETZSCH Proteus^®^ Software—Thermal Analysis—Version 6.1).

#### 2.3.6. Transmission Electron Microscopy Analysis

Transmission electron microscopy (TEM) analysis was performed to observe NLC morphology. Samples were prepared by placing a drop of diluted (1:100) nanoformulations, over a cooper-mesh grid for 2 min, followed by negative staining with uranyl acetate for 30 s. Images were obtained with an accelerating voltage of 80 kV in a JEM-1400 Transmission Electron Microscope (JEOL Ltd., Tokyo, Japan).

### 2.4. Cell Culture Studies

#### 2.4.1. Ethics Statement

The procedures to obtain human biological samples were in accordance to the principles of the Declaration of Helsinki. Human monocytes were isolated from surplus buffy coats from healthy blood donors, kindly donated by Serviço de Imunohemoterapia, Centro Hospitalar de São João (CHSJ) Porto, as part of an agreement covered by the ethical approval of the service, under which blood donors gave written informed consent for the by-products of their blood collections to be used for research purposes (Protocol reference 260/11). No information on age, sex, or any identifying element was provided to the researchers; thus, the samples were analysed anonymously.

#### 2.4.2. Monocyte Isolation and Primary Human Macrophages Differentiation

Monocyte isolation was performed by negative selection, using methods adapted from Oliveira et al. [23]. Briefly, buffy coats from healthy donors were centrifuged for 30 min at 1200× *g*. The peripheral blood mononuclear cell (PBMC) layer was collected and incubated with RosetteSep human monocyte enrichment kit (StemCell Technologies SARL, Grenoble, France) according to the manufacturer’s instructions. The mixture was diluted at a 1:1 ratio with phosphate-buffered saline (PBS) supplemented with 2% of heat-inactivated fetal bovine serum (FBS) (Lonza, Basel, Switzerland) and then layered over Histopaque^®^-1077 (Sigma Aldrich GmbH, Hamburg, Germany) at 1:1:1 ratio. The layered tube was then centrifuged for 20 min at 1200× *g*. The enriched monocyte layer was collected and washed with PBS three times. The washes were performed at 700 g for 20 min to ensure platelet removal. The pellet was re-suspended in culture medium Roswell Park Memorial Institute (RPMI) with glutamax (Invitrogen GmbH, Darmstadt, Germany) supplemented with 10% FBS and 1% penicillin G-streptomycin (Invitrogen GmbH, Darmstadt, Germany). Cells were counted in a Neubauer chamber using the trypan blue exclusion assay (Sigma-Aldrich GmbH, Hamburg, Germany), plated on tissue culture polystyrene plates (BD Biosciences, San Jose, CA, USA), and cultured in a humidified 37 °C, 5% CO_2_ incubator. Monocyte/macrophages differentiation was performed in the presence of 50 ng mL^−1^ of macrophage colony-stimulating factor (M-CSF; Immunotools GmbH, Friesoythe, Germany).

#### 2.4.3. Cell Viability Assessment

Monocytes were treated with RPT-loaded NLCs suspension and corresponding placebos at different concentrations (ranging from 0 to 1750 µg mL^−1^) on the day of isolation. The dilution of NLCs was done in culture medium RPMI with glutamax supplemented with 10% FBS and 1% penicillin G-streptomycin. Upon 24 h of drug treatment, cells were washed three times with warm PBS to remove the excess of nanoparticles that were not internalized by the cells. Cell viability was assessed at days 1 and 7 through the determination of the presence of metabolically active cells using the resazurin reduction assay. This fluorometric method estimates cell viability by measuring the reduction of resazurin into resorufin. For that purpose, 0.01 mg mL^−1^ of resazurin solution (Sigma-Aldrich GmbH, Hamburg, Germany) was added to each well of the black 96-well plate, which was then incubated for three hours in a humidified 37 °C, 5% CO_2_ incubator. Fluorescence intensity was measured with 530 nm excitation wavelength and 590 nm emission wavelength using a Synergy HT Photometer (Biotek Instruments, Winooski, VT, USA). The values were normalized using untreated cells. The percentage of cell viability was calculated by the ratio of the mean of absorbance values from triplicates, between treated and untreated (control) cells for each time-point. The half maximum inhibitory concentration (IC_50_) of each formulation was calculated using a nonlinear regression, with a special dose–response (EC_50_ shift) using the GraphPad Prism6 software program (GraphPad Software Inc., San Diego, CA, USA).

### 2.5. Statistical Analysis

Statistical comparisons were performed using the two-way analysis of variance and differences between groups compared using Tukey’s multiple comparisons test with a *p*-value < 0.05 considered statistically significant. The analyses were performed using the GraphPad Prism6 software program (GraphPad Software Inc., San Diego, CA, USA).

## 3. Results and Discussion

The NLC composition, synthesis method, and the speed and time of sonication for NLC synthesis was chosen according to preliminary formulation studies performed in our group [15,20,22]. After a preliminary screening, three main variables affecting the particle size, EE, and LC were identified: the amount of solid lipid, the amount of liquid lipid, and the amount of surfactant. To model these variables, to evaluate their effect in the selected dependent variables, and to determine the optimum design conditions, a response surface methodology was performed.

Box–Behnken design is one of the main response surface methodologies used for designing experiments. The 3-level, 3-factor design evaluates three independent factors with a fewer number of experiments based on a collection of mathematical and statistical techniques, which leads to reagent savings and to less time-consuming laboratory work [24,25]. In the last decades, the Box–Behnken design has been applied to optimize the process of several drug-loaded nanoparticle formulations [13,14,15,16,17,18].

### 3.1. Selection of the Most Appropriate Lipid Composition

In this study, three different levels of three independent variables (i.e., amount of solid lipid, liquid lipid, and surfactant) were tested, and their effect on three dependent variables (i.e., particle size, EE, and LC) was assessed (Table 2).

A regression analysis using the two-way interaction (linear × quadratic) model, considering a 95% confidence level, was performed. The positive sign of the factors indicates a synergistic effect on the response where the response increases, while the negative sign is indicative of an antagonist effect where the response decreases with the factor [15]. Interaction terms are represented by more than one factor (i.e., *X*_1_*X*_2_, *X*_1_*X*_3_, and *X*_2_*X*_3_) and quadratic relationships are represented by higher-order terms (i.e., *X*_1_^2^, *X*_2_^2^, and *X*_3_^2^). Analyses of variance (ANOVA) for the relevance of the model were performed for the three dependent variables (Appendix A).

According to the analysis of the estimated effects (Table 3), the amount of solid lipid (*X*_1_) had a positive, significant effect (coefficient = 30.833; *p*-value = 0.031) on particle size (*Y*_1_) and a negative, significant effect (coefficient = −0.311; *p*-value = 0.009) on LC (*Y*_3_). This result was expected since increased amount of lipids leads to increased, particle size [16]. In addition, an increasing amount of solid lipid will contribute to an increment in the total lipid amount, which contributes to a lower LC. Accordingly, the amount of liquid lipid (*X*_2_) also had a negative, significant effect (coefficient = −0.345; *p*-value = 0.007) on LC (*Y*_3_).

The amount of surfactant (*X*_3_) had a negative, significant effect (coefficient = −30.167; *p*-value = 0.032) on particle size (*Y*_1_). This result is in agreement with other studies that indicate higher amounts of surfactant promote a decrease in interfacial tension between the lipid and external phase, which will contribute to the formation of smaller particles [15,18]. A significant, positive, quadratic effect of the amount of surfactant (*X*_3_^2^) was observed for EE (coefficient = 3.977; *p*-value = 0.037) and LC (coefficient = 0.105; *p*-value = 0.035). These results suggest that higher amounts of surfactant may promote the formation of more stable nanoparticles that are more able to encapsulate RPT and, thus, increase their LC.

The interaction effects of solid lipid with liquid lipid (*X*_1_*X*_2_) and solid lipid with surfactant (*X*_1_*X*_3_) were not statistically significant for the analysed dependent variables. Nonetheless, the interaction effect of liquid lipid with surfactant (*X*_2_*X*_3_) had a positive, significant effect on EE (coefficient = 6.950; *p*-value = 0.045) and on LC (coefficient = 0.175; *p*-value = 0.046). These results may be explained by the fact that an increase in the amount of liquid lipid and surfactant may produce a less ordered crystalline structure that will create a higher number of cavities to encapsulate RPT, which allow a higher incorporation of RPT inside of the lipid matrix [16].

The value of *R*^2^ was higher than 0.95 (Table 3) for all the regression equations generated, which indicates the validity and significance of this model for the optimization of this nanodelivery system.

Response counter plots were used to graphically show the statistically significant relationship and interaction effects of two independent variables on the dependent variables when a third factor is kept at constant level (Figure 1). The constant level was fixed as the medium level (0) of the correspondent independent variable (Table 1). Figure 1A reveals that simultaneous increase in surfactant and solid lipid had a positive effect (increase) on particle size, promoting the formation of RPT-NLCs with more than 280 nm. Similarly, simultaneous increase in surfactant and liquid lipid had a positive effect (increase) on EE, leading to EE higher than 75% (Figure 1B). Regarding LC, the simultaneous increase in liquid lipid and solid lipid (Figure 1C) and the simultaneous increase in surfactant and liquid lipid (Figure 1D) had a negative effect (decrease) on LC, contributing to the formation of particles with LC values below 2. This effect was more pronounced in the case of the simultaneous increase in liquid lipid and solid lipid (Figure 1C,D).

The prediction and profiling function of STATISTICA 10 (Statsoft Power Solutions, Inc.) software was used to obtain the response desirability profile for the optimized formulation. The factors were settled as particle size in the optimum value of 200 nm, maximum EE, and maximum LC. According to this analysis, the composition of the optimized RPT-loaded NLCs should be 250 mg of solid lipid, 50 mg of liquid lipid, and 80 mg of surfactant (Appendix A).

### 3.2. Validation of the Experimental Design

To confirm the validity of the optimization procedure and to validate the prediction capability of the selected model, three independent replicates were synthetized, and measurements of particle size, EE, and LC were performed. Table 4 shows the predicted and the observed values for these parameters. The observed responses were in good agreement with the predicted values, which validates the selection of this experimental design for a successful optimization of this nanodelivery system.

### 3.3. Characterization of the Optimized Nanoparticles

After validation of the experimental Box–Behnken design, the three replicates and the corresponding placebos were characterized in terms of size, PDI, ζ-potential, EE, and LC (Table 5). The mean hydrodynamic particle sizes of NLCs and RPT-NLCs were 245 ± 4 and 242 ± 9 nm, respectively (Table 5). The incorporation of RPT did not significantly influence the size of nanoparticles. The PDI values were below 0.2 for all formulations (Table 5), which suggests a homogeneous distribution of the developed nanoformulations. The EE of drug in NLCs was 86 ± 4% and their LC was 2.9 ± 0.1 (Table 5). Therefore, NLCs can be considered suitable nanocarriers for the entrapment of RPT.

To assess the effect of drug loading on the lipid matrix of developed nanoformulations, a DSC analysis was performed. DSC thermograms for NLCs and RPT-NLCs were done and the melting point (peak maximum), melting enthalpy (ΔH), and onset, and end ΔH parameters were calculated (Table 6). Results revealed that the addition of RPT promoted a decrease in the analyzed parameters when compared to the placebo nanoformulations. The decreases in the ΔH, melting point, and in the onset temperature were in full agreement with the high entrapment efficacy of RPT in the NLCs. In fact, the presence of the drug is expected to increase the number of defects in the lipid matrix of lipid nanoparticles, creating a disturbance in the crystal order that leads to a decrease of the lipid melting point [17,26].

To evaluate the morphology of the optimized NLCs, TEM was performed. Results show spherical nanoparticles, uniform in shape and diameter, with no visible aggregation (Figure 2). The morphology of NLCs (Figure 2A) and RPT-NLCs (Figure 2B) was similar, which indicates that the incorporation of RPT did not affect the shape of NLCs. TEM analyses revealed mean diameter of particles in the range of 250 nm, validating the above-mentioned results obtained with dynamic light scattering.

### 3.4. Physical Stability of Optimized Nanoparticles

To improve the physical and microbiological stability of the formulations, a lyophilization process was performed. The particle size, PDI, ζ-potential, EE, and LC of lyophilized and liquid suspension nanoparticles were assessed during 3 months (Figure 3). Results revealed no significant changes of particle size, ζ-potential, and EE values within each formulation over time. The PDI values of lyophilized RPT-NLCs had a significant increase when compared to the value obtained after synthesis (T0) (Figure 3B). However, the mean hydrodynamic diameter of these formulations remained stable over time (Figure 3A).

Comparing the same time-point of nanoparticles in suspension with the corresponding lyophilized nanoformulations, increased particle size (Figure 3A) and PDI (Figure 3B) of lyophilized formulations were observed. These results may be explained by the formation of small aggregates that contribute to a higher mean hydrodynamic diameter and to a higher PDI. The ζ-potential values were below −20 mV (Figure 3C), and no tendency to ζ-potential variations was observed, supporting the long-term stability of the developed NLCs. The EE values were higher than 80% (Figure 3D) and stable for all nanoformulations during the 3 months of storage. These results support that the developed nanoparticles are stable both in aqueous suspension and in lyophilized form during at least 3 months at the storage conditions.

### 3.5. Primary Human Macrophage Viability upon Nanoparticles Exposure

To assess the effect of the selected nanoformulations on cell viability, primary human macrophages were exposed to nanoparticles at different concentrations (ranging from 0 to 1750 µg mL^−1^) for 24 h. These studies were performed using primary human macrophages, as RPT-NLCs are intended for pulmonary delivery, and macrophages are a most relevant cell target in the treatment of TB because of their importance as innate and effector cells in the immune response to bacteria, and also because they are the initial cells infected by *Mycobacterium tuberculosis*. Cell viability was measured at days 1 and 7 after nanoparticle exposure and were normalized to the metabolic activity of untreated cells of the same donor, at the same time point. Results revealed that no significant differences were observed between RPT-NLCs and corresponding placebos for either time point. The lower nanoparticle concentrations had no impact on cell viability. The lactate dehydrogenase (LDH) assay was used as indicator of cytotoxicity, and the results of our previous study [19] also revealed that these concentrations of NLCs did not alter the cell membrane integrity. The highest tested concentrations (875 and 1750 µg mL^−1^) impacted viability of primary human macrophages (Figure 4), an effect that was more pronounced at day 7 (Figure 4B). In the presence of 875 µg mL^−1^ of nanoparticles, cell viability was still above 70%, for both analyzed time points, but for the highest concentration used, 1750 µg mL^−1^, macrophage viability was less than 50%; thus, this concentration may be considered toxic for primary human macrophages. These results are in agreement with our previous report on NLC toxicity, in mouse bone marrow-derived macrophages [22]. This effect should be related to the presence of high lipid concentrations, since the effect was observed in both RPT-loaded and non-loaded formulations. The IC_50_ of each formulation was calculated for each time point after incubation with nanoparticles. The values obtained were 1469 µg mL^−1^ for RPT-NLCs and 1540 µg mL^−1^ for NLCs at day 1, and 958 µg mL^−1^ for RPT-NLCs and 1079 µg mL^−1^ for NLCs at day 7. Thus, nanoparticles up to 958 µg mL^−1^ are suitable to be tested in in vitro studies using primary human macrophages. The inhibitory activities (MICs) of RPT against *Mycobacterium tuberculosis* residing in human monocyte-derived macrophages are from 0.015 (intracellular bacteria) to 0.06 μg mL^−1^ (extracellular bacteria) [27]. Accordingly, nontoxic concentrations of RPT-NLCs (i.e., 950 µg mL^−1^) have in their composition enough concentration of RPT (i.e., 25 µg mL^−1^) to inhibit more than 99% of the bacterial population. To further confirm RPT-NLCs activities, in terms of drug efficiency, future studies will be performed using *Mycobacterium tuberculosis*-infected cells.

## 4. Conclusions

The present work involves the optimization of NLCs loaded with RPT, a rifamycin derivative that has been made to improve the antimycobacterial activity of other first-line anti-TB drugs. RPT-NLCs were developed, and their size, EE, and LC were successfully optimized using a quality-by-design approach. Based on the Box–Behnken results, the amount of liquid lipid and surfactant were the most determinant factors that influenced the physical properties of RPT-NLCs. The optimized RPT-NLCs had a mean diameter of 242 ± 9 nm, with a homogeneous distribution of particles (PDI < 0.2), and a highly negative surface charge (ζ-potential = −22 ± 2 mV). The EE values were higher than 80%, and DSC analysis enabled the confirmation of the efficient encapsulation of RPT. These results suggest that these nanoformulations could potentially be exploited as a delivery system with improved drug EE. TEM analysis revealed spherical nanoparticles, uniform in shape and width, with no visible aggregation, validating the previous physical characterization measurements. Stability studies indicated that RPT-NLCs and the corresponding placebos were stable over time, and no toxicity was observed in primary human macrophages viability for RPT-NLCs up to 1000 μg mL^−1^, which is indicative of their safe application. Overall, the developed nanoparticles show interesting properties for a new delivery form of RPT. Future studies should include the use of more complex in vitro characterization techniques, and *in vivo* efficacy studies in clinically relevant models, to evaluate the application of this new RPT delivery system in TB therapy.

## Figures and Tables

**Figure 1 pharmaceutics-12-00075-f001:**
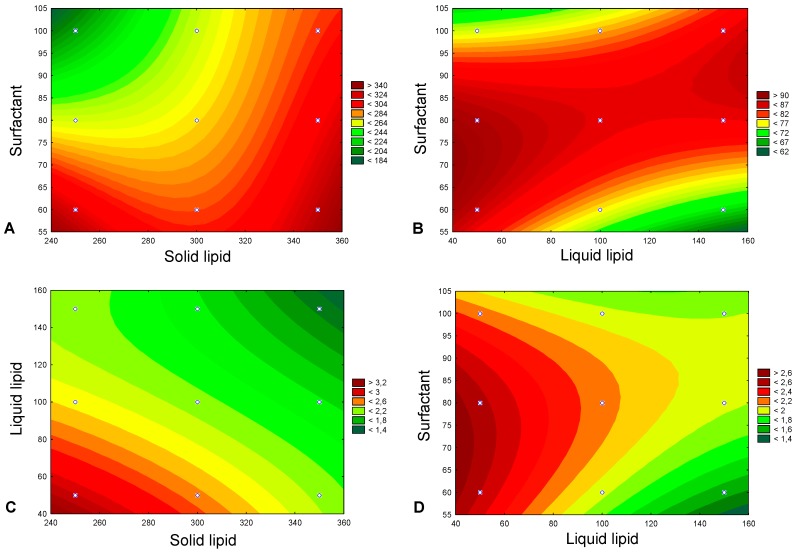
Response contour plots, representing the statistically significant effects (*p*-value < 0.05), namely (**A**) the effect of the amount of solid lipid (*X*_1_) and amount of surfactant (*X*_3_) on particle size (*Y*_1_), (**B**) the effect of the amount of liquid lipid (*X*_2_) and amount of surfactant (*X*_3_) on encapsulation efficiency (*Y*_2_), (**C**) the effect of the amount of solid lipid (*X*_1_) and amount of liquid lipid (*X*_2_) on loading capacity (*Y*_3_), and (**D**) the effect of the amount of liquid lipid (*X*_2_) and amount of surfactant (*X*_3_) on loading capacity (*Y*_3_).

**Figure 2 pharmaceutics-12-00075-f002:**
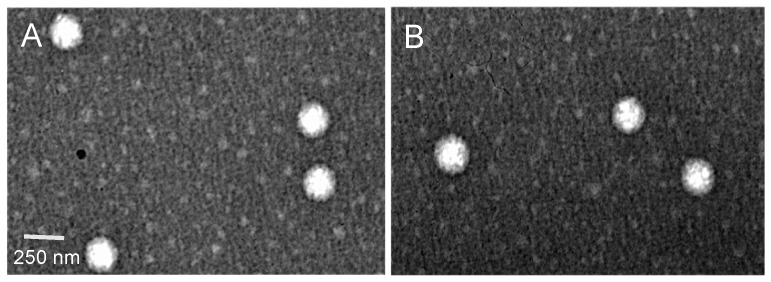
Transmission electron microscopy images of (**A**) NLCs and (**B**) RPT-loaded NLCs, at 50,000× magnification. The white bar represents 250 nm.

**Figure 3 pharmaceutics-12-00075-f003:**
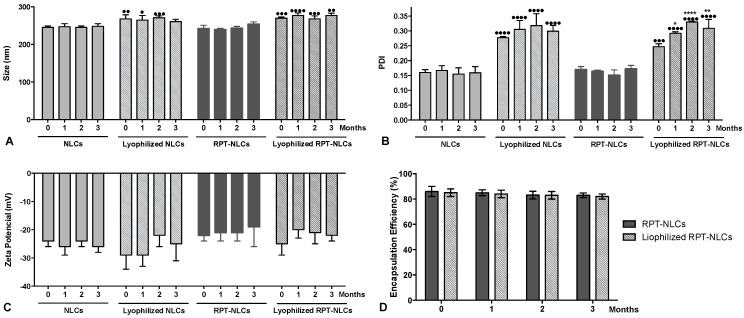
Storage stability of (**A**) particle size, (**B**) polydispersity index (PDI), (**C**) zeta potential, and (**D**) encapsulation efficiency of the lyophilized and liquid suspension nanoparticles during 3 months of storage at 20 °C. Data expressed as mean ± SD (n = 3). Statistical comparisons of the means were performed using the two-way analysis of variance and differences between groups compared using Tukey’s multiple comparisons test. * *p* < 0.05; ** *p* < 0.01; **** *p* < 0.0001 compared with the corresponding T0 (measurement after synthesis). • *p* < 0.05; •• *p* < 0.01; ••• *p* < 0.001; •••• *p* < 0.0001 compared with the corresponding placebo at the same time-point.

**Figure 4 pharmaceutics-12-00075-f004:**
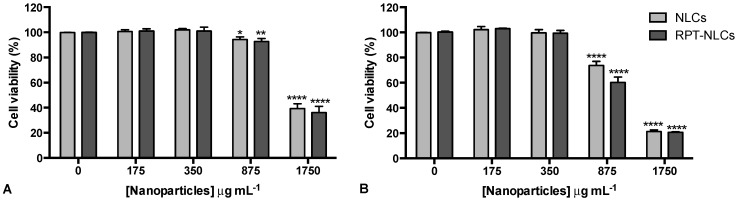
Primary human macrophage viability. Cells were exposed to RPT-NLCs and corresponding placebos at different concentrations (ranging from 0 to 1750 µg mL^−1^). Cell viability was measured using the resazurin reduction assay after (**A**) 1 d and (**B**) 7 d. Metabolic activity was normalized to untreated cells of the same donor and time point. Data are expressed as mean ± standard deviation (n = 3). Statistical comparisons of the means were performed using the two-way analysis of variance and differences between groups compared using Tukey’s multiple comparisons test. * *p* < 0.05, ** *p* < 0.01, **** *p* < 0.0001 compared to untreated cells.

**Table 1 pharmaceutics-12-00075-t001:** Independent and dependent variables of the 3-level, 3-factor Box–Behnken design.

Independent Variables	Coded Levels		
	Low Level (−1)	Medium Level (0)	High Level (1)
*X*_1_ = solid lipid (mg)	250	300	350
*X*_2_ = liquid lipid (mg)	50	100	150
*X*_3_ = surfactant (mg)	60	80	100
**Dependent variables**		**Criteria**	
*Y*_1_ = particle size (nm) *Y*_2_ = encapsulation efficiency (%) *Y*_3_ = loading capacity		Optimum (200 nm)	
		Maximum (100%)	
		Maximum	

**Table 2 pharmaceutics-12-00075-t002:** Observed responses in Box–Behnken design for rifapentine (RPT)-loaded nanostructured lipid carriers (NLCs).

Sample	Independent Variables			Dependent Variables		
	*X*_1_ (mg)	*X*_2_ (mg)	*X*_3_ (mg)	*Y*_1_ (nm)	*Y*_2_ (%)	*Y* _3_
1	250	50	80	235	90.3	3.01
2	350	50	80	311	86.8	2.17
3	250	150	80	275	85.8	2.15
4	350	150	80	334	78.7	1.57
5	250	100	60	317	75.5	2.16
6	350	100	60	330	86.6	1.92
7	250	100	100	207	75.0	2.14
8	350	100	100	294	67.2	1.49
9	300	50	60	271	87.4	2.50
10	300	150	60	322	69.2	1.54
11	300	50	100	251	75.2	2.15
12	300	150	100	272	84.8	1.89
13	300	100	80	277	84.9	2.12
14	300	100	80	254	89.1	2.23
15	300	100	80	282	83.2	2.08

The independent variables were the amounts of solid lipid (*X*_1_), liquid lipid (*X*_2_), and surfactant (*X*_3_), while mean hydrodynamic particle size (*Y*_1_), encapsulation efficiency (*Y*_2_), and loading capacity (*Y*_3_) were the dependent variables in study.

**Table 3 pharmaceutics-12-00075-t003:** Regression analysis for particle size (*Y*_1_), encapsulation efficiency (EE) (*Y*_2_), and loading capacity (LC) (*Y*_3_) using the two-way interaction model (linear vs quadratic) based on the effect of the amount of solid lipid (*X*_1_), liquid lipid (*X*_2_), and surfactant (*X*_3_).

	Size-*Y*_1_		EE-*Y*_2_		LC-*Y*_3_	
	Coeff.	*p*-Value	Coeff.	*p*-Value	Coeff.	*p*-Value
**Intercept**	**284.917**	**0.000**	**80.208**	**0.000**	**2.058**	**0.000**
***X*_1_**	**30.833**	**0.031**	−1.492	0.318	**−0.311**	**0.009**
***X*_1_^2^**	−6.438	0.239	0.852	0.394	0.003	0.906
***X*_2_**	16.500	0.097	−2.817	0.131	**−0.345**	**0.007**
***X*_2_^2^**	−2.438	0.595	−0.685	0.477	−0.044	0.164
***X*_3_**	**−30.167**	**0.032**	−3.033	0.116	−0.075	0.122
***X*_3_^2^**	−1.563	0.727	**3.977**	**0.037**	**0.105**	**0.035**
***X*_1_** ***X*_2_**	−4.250	0.627	−0.900	0.613	0.065	0.236
***X*_1_** ***X*_3_**	18.500	0.132	−4.725	0.090	−0.103	0.119
***X*_2_** ***X*_3_**	−7.500	0.421	**6.950**	**0.045**	**0.175**	**0.046**
***R*^2^**	0.976		0.975		0.994	

Interaction terms are represented by more than one factor (i.e., *X*_1_*X*_2_, *X*_1_*X*_3_, and *X*_2_*X*_3_) and quadratic relationships are represented by higher-order terms (i.e., *X*_1_^2^, *X*_2_^2^, and *X*_3_^2^). Statistically significant parameters (*p*-value < 0.05 with a 95% confidence interval) are highlighted in bold.

**Table 4 pharmaceutics-12-00075-t004:** Validation of the predicted optimal results with experimental values. (n = 3).

Dependent Variables	Predicted Values	Experimental Values
*Y*_1_ = particle size (nm)	235	242 ± 9
*Y*_2_ = EE (%)	90	86 ± 4
*Y*_3_ = LC	3.0	2.9 ± 0.1

**Table 5 pharmaceutics-12-00075-t005:** Characterization of the optimized RPT-loaded NLCs and corresponding placebos (n = 3) in terms of mean hydrodynamic particle size, polydispersity index, zeta potential, encapsulation efficiency, and loading capacity.

Samples	Diameter (nm)	PDI	ζ-Potential (mV)	EE (%)	LC
NLCs	245 ± 4	0.16 ± 0.01	–24 ± 2	-	-
RPT-NLCs	242 ± 9	0.17 ± 0.01	–22 ± 2	86 ± 4	2.9 ± 0.1

Data expressed as mean ± SD (n = 3).

**Table 6 pharmaceutics-12-00075-t006:** Differential scanning calorimetry (DSC) parameters of NLCs and RPT-NLCs: melting enthalpy (ΔH), the onset, the melting point (peak maximum), and the end parameters.

Samples	ΔH (Jg^−1^)	ΔT_onset_ (°C)	Melting Point (°C)	ΔT_end_ (°C)
NLCs	93.7	58.7	60.3	61.8
RPT-NLCs	89.0	49.0	58.7	60.5

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
