# Peer review of "Optimization of Rifapentine-Loaded Lipid Nanoparticles Using a Quality-by-Design Strategy"

_pharmaceutics, 2020, doi:10.3390/pharmaceutics12010075_

Round 1
Reviewer 1 Report
In this manuscript, Magalhaes J R and coworkers describe the generation and characterization of a lipid nanoparticle delivery system, namely nanostructured lipid carriers (NLC) loaded with the anti-microbial drug rifapentine (RPT) for tuberculosis (TB) therapy. The topic in and of itself is an interesting one and the observations presented in the study are interesting. The methodology used is appropriate and results obtained are good. Regarding Methodology, the set of experiments based on the box-beknhen design conducted to design the most optimized lipid nanoparticles are an elegant contribution to Nanotechnology field. However, some concerns have to be taken into consideration.
Authors analyzed the toxicity of NLC loaded with RPT (NLP-RPT) using a MTT-type assay using resazurin. However, methods based on the metabolic capacity of viable cells do not measure cell death but inhibition of proliferation. Since this kind of assays does not always indicate that cells are dying, authors should complement their analysis by using other techniques to evaluate cell death in macrophages (i.e. flow cytometry staining with annexin V and PI).
Following with toxicity assays, which NLC-RTP have been used in macrophage viability assays?, “freshly” generated NLCs or lyophilized NLCs? It would be interesting that these experiments were carried out using lyophilized NLCs since, as expected, they should be used in future in vivo studies on clinically relevant models. Auhtors should claryfy this point.
Finally, regarding cytotoxicity experiments, authors did not carry out any experiment to analyze the bioactivity of their NLP-RPT in TB. As authors claim in Conclusions: “Future studies should include the use of more complex in vitro characterization techniques, and in vivo efficacy studies in clinically relevant models, to evaluate the application of this new RPT delivery system in TB therapy”. This reviewer understands that authors cannot perform that set of experiments for this work; however, it would be desirable that authors included some preliminary experimentation of the bioactivity of NLP-RPT against TB (i.e. cytotoxicity assay against TB-infected macrophages or mycobacterium tuberculosis) as “proof of concept” of this lipid nanoparticles loaded with RTP. If that were at all possible, it would greatly improve the manuscript.
In the study of the physical stability of NLC-RTPs, a significant increase of PDI of lyophilized formulation was observed that authors attribute to formation of small aggregates that contribute to higher mean hydrodynamic diameter and to a higher PDI. These “small” aggregates could explain the slight increase in lipid nanoparticle diameter (Figure 3A), but it is a little harder to understand in case of PDI since the increase observed is more than twice (Figure 3B). Authors should give a more detailed explanation on this point. In this regard, this apparent loss of homogeneity of NLC-RTPs after lyophilization would affect their bioactivity?
Specific points:
Acronyms should be defined the first time used in the manuscript (i.e. EE and LC in page 2, PDI in page 5, DSC TEM in page 6). Authors should be revised this point. In page 5, temperature used during lyophilization process was -800ºC. It should be revised. Figure 1 should be explained in more detail in the Results and Discussion section (page 11) since this kind of graphics cannot be understood by non-expert readers in this topic. In this line, side legend showed at the right side of each graphic should also be explained. Finally, also in Figure 1, axis labels should be showed in a larger font.
Author Response
Manuscript ID: pharmaceutics-675404
Title: Optimization of lipid nanoparticles for tuberculosis therapy based on a quality-by-design strategy
The Manuscript pharmaceutics-675404, entitled “Optimization of lipid nanoparticles for tuberculosis therapy based on a quality-by-design strategy” was carefully rewritten to address all the concerns and suggestions raised by the Reviewers. We believe that the comments added to the text are now in agreement with the considerations made by the Reviewers. A more detailed explanation of the improvements made is described below. In addition, the changes made were highlighted in the manuscript by using the track changes mode in MS word (please see the attachment).
Comments to the Author
In this manuscript, Magalhaes J R and coworkers describe the generation and characterization of a lipid nanoparticle delivery system, namely nanostructured lipid carriers (NLC) loaded with the anti-microbial drug rifapentine (RPT) for tuberculosis (TB) therapy. The topic in and of itself is an interesting one and the observations presented in the study are interesting. The methodology used is appropriate and results obtained are good. Regarding Methodology, the set of experiments based on the box-beknhen design conducted to design the most optimized lipid nanoparticles are an elegant contribution to Nanotechnology field. However, some concerns have to be taken into consideration.
Authors analyzed the toxicity of NLC loaded with RPT (NLP-RPT) using a MTT-type assay using resazurin. However, methods based on the metabolic capacity of viable cells do not measure cell death but inhibition of proliferation. Since this kind of assays does not always indicate that cells are dying, authors should complement their analysis by using other techniques to evaluate cell death in macrophages (i.e. flow cytometry staining with annexin V and PI).
Author response: We understand the Reviewer’s concern. The resazurin assay estimates cell viability by measuring the reduction of resazurin into resorufin by aerobic respiration of metabolically active cells. As the Reviewer mentioned, this fluorometric method is not indicative that cells are dying and other techniques should be used to complement this analysis. In fact, the Lactate Dehydrogenase (LDH) assay was used as indicator of cytotoxicity. This method is based on the release of LDH into the medium due to plasma membrane permeabilization as a result of cell death. These results were in agreement with the ones of the resazurin assay, and to clarify this point, new information was added to “Results and Discussion” section: “The lactate dehydrogenase (LDH) assay was used as indicator of citotoxicity and the results of our previous study [19] also revealed that these concentrations of NLCs did not alter the cell membrane integrity.” The list of references was updated accordingly.
Following with toxicity assays, which NLC-RTP have been used in macrophage viability assays?, “freshly” generated NLCs or lyophilized NLCs? It would be interesting that these experiments were carried out using lyophilized NLCs since, as expected, they should be used in future in vivo studies on clinically relevant models. Authors should claryfy this point.
Author response: Macrophage viability assays were performed using “freshly” generated NLCs (in suspension). We understand the concern of the Reviewer but in fact, we would like to exploit the inhalatory route of administration and recent in vivo studies have been tested the biocompatibility, clearance and biodistribution of nanoparticles by the intranasal administration or via oropharyngeal aspiration of nanoparticles in suspension (i.e. Woods et al., 2015; Patel et al., 2016). We totally agree that we should clarify this point thus, new information was added to section 2.4.3: “Monocytes were treated with RPT-loaded NLCs suspension and corresponding placebos at different concentrations (ranging from 0 to 1750 mg mL-1) on the day of isolation”.
Supporting References:
Woods A, Patel A, Spina D, Riffo-Vasquez Y, Babin-Morgan A, De Rosales RT, Sunassee K, Clark S, Collins H, Bruce K, Dailey LA, Forbes B. In vivo biocompatibility, clearance, and biodistribution of albumin vehicles for pulmonary drug delivery. Journal of Controlled Release 210, 1-9 (2015).
Patel A, Woods A, Riffo-Vasquez Y, Babin-Morgan A, Jones MC, Jones S, Sunassee K, Clark S, R TMDR, Page C, Spina D, Forbes B, Dailey LA. Lung inflammation does not affect the clearance kinetics of lipid nanocapsules following pulmonary administration. Journal of Controlled Release 235, 24-33 (2016).
Finally, regarding cytotoxicity experiments, authors did not carry out any experiment to analyze the bioactivity of their NLP-RPT in TB. As authors claim in Conclusions: “Future studies should include the use of more complex in vitro characterization techniques, and in vivo efficacy studies in clinically relevant models, to evaluate the application of this new RPT delivery system in TB therapy”. This reviewer understands that authors cannot perform that set of experiments for this work; however, it would be desirable that authors included some preliminary experimentation of the bioactivity of NLP-RPT against TB (i.e. cytotoxicity assay against TB-infected macrophages or mycobacterium tuberculosis) as “proof of concept” of this lipid nanoparticles loaded with RTP. If that were at all possible, it would greatly improve the manuscript.
Author response: We are grateful to the Reviewer for the valuable suggestion. In fact, in this work we were mainly focused in the optimization of the formulation in terms of the desired characteristics of the final formulation. In addition, we performed several sets of experiments to characterize the formulation, being this manuscript a systematic work in the pharmaceutical sciences field of an optimization and characterization of one novel nanotechnology system to delivery an already known, and effective, drug (i.e. rifapentine). Regarding the experiments with cells, our main goal was to prove that our system is biocompatible but in the future, we will characterize the in vitro antimycobacterial efficacy using Mycobacterium tuberculosis-infected cells,.and we will perform relevant in vivo experiments.
In the study of the physical stability of NLC-RTPs, a significant increase of PDI of lyophilized formulation was observed that authors attribute to formation of small aggregates that contribute to higher mean hydrodynamic diameter and to a higher PDI. These “small” aggregates could explain the slight increase in lipid nanoparticle diameter (Figure 3A), but it is a little harder to understand in case of PDI since the increase observed is more than twice (Figure 3B). Authors should give a more detailed explanation on this point. In this regard, this apparent loss of homogeneity of NLC-RTPs after lyophilization would affect their bioactivity?
Author response: The authors acknowledge the comment of the Referee. In fact, polydispersity index (PDI) was used to describe the degree of uniformity of the size distribution of the developed NLCs. PDI is a dimensionless number calculated from a two-parameter fit to the correlation data (the cumulants analysis). The numerical value of PDI ranges from 0 (perfectly uniform sample in terms of size distribution) to 1 (highly polydisperse sample with multiple particle size populations). For drug delivery systems using lipid-based carriers, values of 0.3 and below are considered acceptable and indicative of homogeneous populations (Danaei 2018). After synthesis, all the NLCs developed (liquid suspension and lyophilized formulations) have PDI values below 0.3, acceptable for drug delivery applications. However, as the Reviewer mentioned, PDI of lyophilized NLCs is increased when compared to liquid suspension NLCs, being this phenomena frequently noticed by several authors after the lyophilization process. The lyophilization process used in the present study involved the use of aerosil as cryoprotectant. The type of the sugar and the ratio sugar:lipid nanoparticles are critical factors that interfered with PDI and size distributions. However, the bioactivity of the NLCs is not expected to be affected since the amount of entrapped drug, translated by the %EE of the lyophilized formulations was not affected by the lyophilization process.
Supporting reference: Danaei M, Dehghankhold M, Ataei S, Davarani FH, Javanmard R, Dokhani A, Khorasani S, Mozafari MR. Impact of Particle Size and Polydispersity Index on the Clinical Applications of Lipidic
Nanocarrier Systems. Pharmaceutics 10, 57 (2018).
Specific points:
Acronyms should be defined the first time used in the manuscript (i.e. EE and LC in page 2, PDI in page 5, DSC TEM in page 6). Authors should be revised this point. In page 5, temperature used during lyophilization process was -800ºC. It should be revised.Author response: The acronyms were correctly defined and the temperature used during lyophilization was corrected to -80ºC.
Figure 1 should be explained in more detail in the Results and Discussion section (page 11) since this kind of graphics cannot be understood by non-expert readers in this topic. In this line, side legend showed at the right side of each graphic should also be explained. Finally, also in Figure 1, axis labels should be showed in a larger font.
Author response: We agree with the reviewer’s suggestion. Thus, new information was added to the “Results and Discussion” section and Figure 1 was modified for a better understanding of the data.
New information inserted in section 3: “Figure 1A reveals that simultaneous increase in surfactant and solid lipid has a positive effect (increase) on particle size, promoting the formation of RPT-NLCs with more than 280 nm. Similarly, simultaneous increase in surfactant and liquid lipid has a positive effect (increase) on EE, leading to EE higher than 75% (Figure 1B). Regarding LC, the simultaneous increase in liquid lipid and solid lipid (Figure 1C) and the simultaneous increase in surfactant and liquid lipid (Figure 1D) have a negative effect (decrease) on LC, contributing for the formation of particles with LC values below 2. This effect is more pronounced in the case of the simultaneous increase in liquid lipid and solid lipid (Figure 1C and 1D).” Figure 1 was modified.

Reviewer 2 Report
The authors have carried out an interesting experimental design to optimize the nanoparticle composition using the minimum experiments. They have obtained nanoparticles with suitable properties to be used as a new delivery form of RPT.
Nonetheless, this manuscript does not explain why this formulation is better than the free drug. The authors suggest that incorporation of RPT into NLC improve pharmacokinetics and pharmacodynamics properties of the treatment, but they do not demonstrate any of the advantages of using NLC. This work is far from demonstrating the applicability of the formulation in tuberculosis therapy. Therefore, the title of the manuscript is too ambitious and the authors must rewrite it.
MAYOR COMMENTS:
Section 2.4. The authors indicate that RPT triggers acute renal failure but instead of using renal cell cultures they determine cytotoxicity of RPT-NLC in pulmonary macrophages arguing that this is the target cell of the pathogen. Are the authors thinking about a pulmonary administration of nanoparticles? If the final goal is a systemic administration of the treatment, a renal cell culture would have given much more information about the cytotoxicity of the treatment. The selection of cell type should be deeper explained.
Besides, the authors should indicate which is the medium used for the dilution of NLC to apply them into cell culture. Does this medium contain FBS and M-CSF? These factors can prevent cells from entering apoptosis and therefore it is important to determine the composition of cell medium.
Taking into account that this manuscript does not demonstrate the efficiency of the formulations, the authors should at least indicate the IC50 of the drug activity (in terms of drug efficiency) and explain whether the required concentration of NLC to obtain this concentration of drug is below the CC50 of the NLCs (in terms of cytotoxicity).
Finally, the authors should explain why the NLC are washed after 24 h.
Section 2.3.2. Taking into account the low solubility of the drug in water, the authors should add controls in order to demonstrate that the non-entrapped drug is in the aqueous solution. For instance, by repeating the same process without adding lipids or surfactants in order to quantify the maximum amount of drug that can be recovered from this assay.
MINOR COMMENTS
Section 2.1. In lines 82-83, the authors indicate that “RPT-loaded NLCs were prepared adding 2.6% w/w of RPT to the total lipid phase”. In lines 95-96 they say that “the amount of drug was set at fixed level”. It is ambiguous because the total amount of lipids change from one formulation to others, so the amount of drug should change as well in order to maintain the percentage. Table 2. According to these data, samples 13-15 are composed of same amounts of solid lipid, liquid lipid and surfactant but obtained values of dependent variables are different. Please check for possible mistakes in the composition of the samples. Otherwise, the authors should explain or comment this variability. Conclusions. Lines 423-428 should not be in this section, they are more suitable for other sections like the introduction. Other considerations:
- Misspellings: line 188: macrophages cultures, line 198: ressuspended, line 395: 24h.
- Instrument description in materials and methods section is not homogeneous. Sometimes instrumentation is described by indicating the country, sometimes the city is also added (TEM analysis) sometimes none of them (DSC analysis).
Author Response
Title: Optimization of lipid nanoparticles for tuberculosis therapy based on a quality-by-design strategy
The Manuscript ID pharmaceutics-675404, entitled “Optimization of lipid nanoparticles for tuberculosis therapy based on a quality-by-design strategy” was carefully rewritten to address all the concerns and suggestions raised by the Reviewers. We believe that the comments added to the text are now in agreement with the considerations made by the Reviewers. A more detailed explanation of the improvements made is described below. In addition, the changes made were highlighted in the manuscript by using the track changes mode in MS word (please see the attachment).
Comments to the Author
The authors have carried out an interesting experimental design to optimize the nanoparticle composition using the minimum experiments. They have obtained nanoparticles with suitable properties to be used as a new delivery form of RPT. Nonetheless, this manuscript does not explain why this formulation is better than the free drug. The authors suggest that incorporation of RPT into NLC improve pharmacokinetics and pharmacodynamics properties of the treatment, but they do not demonstrate any of the advantages of using NLC. This work is far from demonstrating the applicability of the formulation in tuberculosis therapy. Therefore, the title of the manuscript is too ambitious and the authors must rewrite it.
Author response: We are grateful for the reviewer’s suggestion. Although the application of these nanoparticles was intended from the design phase, and still is to develop an alternative delivery strategy for the anti-tuberculosis drug rifapentine (RPT), we agree that within the current study we have not addressed all the necessary steps for that application. Thus, the title of the manuscript was changed to: “ Optimization of rifapentine-loaded lipid nanoparticles using a quality-by-design strategy”.
MAYOR COMMENTS:
1. Section 2.4. The authors indicate that RPT triggers acute renal failure but instead of using renal cell cultures they determine cytotoxicity of RPT-NLC in pulmonary macrophages arguing that this is the target cell of the pathogen. Are the authors thinking about a pulmonary administration of nanoparticles? If the final goal is a systemic administration of the treatment, a renal cell culture would have given much more information about the cytotoxicity of the treatment. The selection of cell type should be deeper explained.
Author response: We understand the Reviewer’s concern. In fact, rifapentine (RPT) is generally administered through the oral route and one of its major adverse effects is acute renal failure. In our work, RPT-loaded NLCs were design not only to carry and protect RPT but also to efficiently deliver RPT to the infection site. As it is well described in literature, tuberculosis occurs due to the inhalation of infectious aerosols of Mycobacterium tuberculosis and bacilli colonization of alveolar macrophages. Macrophages are a most relevant cell target in the treatment of tuberculosis since they are the initial cells infected by Mycobacterium tuberculosis and also important innate and effector cells in the immune response to bacteria. Accordingly, we used primary human macrophages to test the cytotoxicity of our developed formulations. In addition, we would like to exploit the pulmonary administration of RPT-loaded NLCs since the lungs are the main sites of tuberculosis infection. This approach may contribute to reduce the amount and frequency of dosage, and thereby minimizing dose-dependent side effects of free drug administration. To better clarify the selection of the cell type, new information was added to section 3.5: “To assess the effect of the selected nanoformulations on cell viability, primary human macrophages were exposed to nanoparticles at different concentrations (ranging from 0 to 1750 mg ml-1) for 24 h. These studies were performed using primary human macrophages as RPT-NLC are intended for pulmonary delivery, and macrophages are a most relevant cell target in the treatment of tuberculosis due to their importance as innate and effector cells in the immune response to bacteria and also because they are the initial cells infected by Mycobacterium tuberculosis.” Moreover, acute renal failure in response to the drug is a complex process related with the drug administration route, frequently associated with the administration of other drugs and often in people bearing other risk factors that make them more susceptible to this serious side-effect. Since acute renal failure is a complex phenomena the replication of this in renal cells would be extremely reductionist, and not accurate, since this effect must be studied in all organ (kidney and not in one specific type of cell), with the metabolites of rifapentine (and not with the single drug), and including the systemic effect (with a increase in the creatinine clearance), a decrease of the plasma creatine and a decrease of the TFG. This important suggestion will be taken into account in the design of future studies, including pre-clinical in vivo and clinical experiments.
2. Besides, the authors should indicate which is the medium used for the dilution of NLC to apply them into cell culture. Does this medium contain FBS and M-CSF? These factors can prevent cells from entering apoptosis and therefore it is important to determine the composition of cell medium.
Author response: The medium used for the dilution of NLCs was the same medium used in primary human macrophages cultures, namely RPMI with glutamax, supplemented with 10% FBS and 1% penicillin G-streptomycin. To better clarify the procedure, new information was added to section 2.4.3: “The dilution of NLCs was done in culture medium RPMI with glutamax, supplemented with 10% FBS and 1% penicillin G-streptomycin.”
3. Taking into account that this manuscript does not demonstrate the efficiency of the formulations, the authors should at least indicate the IC50 of the drug activity (in terms of drug efficiency) and explain whether the required concentration of NLC to obtain this concentration of drug is below the CC50 of the NLCs (in terms of cytotoxicity).
Author response: The inhibitory activity (MIC) of rifapentine (RPT) against Mycobacterium tuberculosis residing in human monocyte-derived macrophages was determined by Mor and colleagues (1995). In that study, the MIC was defined as the lowest drug concentration that inhibited more than 99% of the bacterial population within 8 days of observation. Results revealed that for RPT, the MICs were from 0.015 (intracellular bacteria) to 0.06 μgmL-1 (extracellular bacteria) (Mor, 1995). Taking in consideration those results, and based on the IC50 of our developed formulations, non-toxic concentrations of RPT-NLCs (i.e. 950 mg mL-1) have in their composition enough concentration of RPT (i.e. 25 μgmL-1) to inhibit more than 99% of the bacterial population. To further confirm, future studies will include the evaluation of RPT and RPT-NLCs activities, in terms of drug efficiency, using Mycobacterium tuberculosis-infected cells.
New information was added to section 3.5: “The inhibitory activities (MICs) of RPT against Mycobacterium tuberculosis residing in human monocyte-derived macrophages are from 0.015 (intracellular bacteria) to 0.06 μgmL-1 (extracellular bacteria) [27]. Accordingly, non-toxic concentrations of RPT-NLCs (i.e. 950 mg mL-1) have in their composition enough concentration of RPT (i.e. 25 mg mL-1) to inhibit more than 99% of the bacterial population. To further confirm RPT-NLCs activities, in terms of drug efficiency, future studies will be performed using Mycobacterium tuberculosis-infected cells.”
Supporting References:
Mor N, Simon B, Mezo N, Heifets L. Comparison of Activities of Rifapentine and Rifampin against Mycobacterium tuberculosis Residing in Human Macrophages. Antimicrobial agents and chemotherapy 2073-2077 (1995).
4. Finally, the authors should explain why the NLC are washed after 24 h.
Author response: After 24 h of NLCs incubation, cells were washed to remove the excess of nanoparticles that were not internalized by the cells. To clarify the procedure, new information was added to section 2.4.3: “Upon 24 h of drug treatment, cells were washed three times with warm PBS, to remove the excess of nanoparticles that were not internalized by the cells”.
5. Section 2.3.2. Taking into account the low solubility of the drug in water, the authors should add controls in order to demonstrate that the non-entrapped drug is in the aqueous solution. For instance, by repeating the same process without adding lipids or surfactants in order to quantify the maximum amount of drug that can be recovered from this assay.
Author response: We are grateful for the Reviewer’s suggestion. In fact, rifapentine has low solubility in water (0,0213 mg/mL). This low solubility in water and its lipophilic profile can be seen as an advantage for drug encapsulation in lipid nanoparticles. In our work, the optimized formulations included the encapsulation of 10 mg of rifapentine in 6 mL of water (maximum concentration of 1,67 mg/mL). RPT-loaded NLCs were diluted 40x before being transferred to the Ultrafree® Centrifugal Filter Device. Thus, assuming that there was 100% of non-entrapped drug, the maximum concentration of rifapentine would be 0,042 mg/mL being above its solubility in water. We did not check the maximum amount of drug that can be recovered from this assay but we did collect the solution on the upper part of the filter, and after destroying the particles with acetonitrile, the amount of drug was quantify. The obtained results were complementary and in agreement with the ones obtained by the method described in section 2.3.2. This supports that rifapentine should be inside the lipid matrix of NLCs.
MINOR COMMENTS
1. Section 2.1. In lines 82-83, the authors indicate that “RPT-loaded NLCs were prepared adding 2.6% w/w of RPT to the total lipid phase”. In lines 95-96 they say that “the amount of drug was set at fixed level”. It is ambiguous because the total amount of lipids change from one formulation to others, so the amount of drug should change as well in order to maintain the percentage.
Author response: We totally agree that a clarification should be made. The 2,6% (w/w) referred in section 2.1.1 were calculated taking in consideration the total amount of lipids and surfactant of the optimized formulation given by the box-behnken design (250 mg of solid lipid, 50 mg of liquid lipid and 80 mg of surfactant; total amount = 380 mg). This information is ambiguous thus, section 2.1.1 was corrected to: “RPT-loaded NLCs were prepared using the above-mentioned approach but adding 10 mg of RPT (Sigma-Aldrich, MA, USA) to the lipid phase before melting.” To better clarify the procedure, we also add new information on section 2.2.1: “ The amount of drug (10 mg), sonication amplitude (70%), sonication time (5 min) and final volume (6 mL) were set at fixed levels.”
2. Table 2. According to these data, samples 13-15 are composed of same amounts of solid lipid, liquid lipid and surfactant but obtained values of dependent variables are different. Please check for possible mistakes in the composition of the samples. Otherwise, the authors should explain or comment this variability.
Author response: Samples 13-15 have exactly the same composition and these repetitions of the central point were done for estimation of the experimental error. Indeed, the algorithm used by the box-behnken design takes in consideration this variability to calculate the effect of the independent variables on the dependent variables responses.
3. Lines 423-428 should not be in this section, they are more suitable for other sections like the introduction.
Author response: We are grateful for the Reviewer’s suggestion. The information of lines 423-428 was deleted from conclusions and added to the Introduction section: “In 2018, there were half a million new cases of rifampicin-resistant TB of which 78% had multidrug-resistant TB [1]. Despite significant technological innovations introduced in the last 10 years, the multidrug-resistance TB crisis, undetected or not notified TB cases, a suboptimal response to the TB and HIV co-epidemics, the high costs for TB patients and the slow uptake of new effective tools constitute persistent and serious challenges in tackling TB epidemic [1,2,6]. This fact encourages the development of new treatment regimens including other anti-TB drugs.”
The list of References was updated accordingly.
Other considerations:
1. Misspellings: line 188: macrophages cultures, line 198: ressuspended, line 395: 24h.
Author response: The misspellings were revised.
2. Instrument description in materials and methods section is not homogeneous. Sometimes instrumentation is described by indicating the country, sometimes the city is also added (TEM analysis) sometimes none of them (DSC analysis).
Author response: We totally agree with the reviewer’s comment. Thus, the instrument descriptions were modified.
